# Interpenetrating Composites: A Nomenclature Dilemma

**DOI:** 10.3390/ma18020273

**Published:** 2025-01-09

**Authors:** Konda Gokuldoss Prashanth

**Affiliations:** 1Department of Mechanical and Industrial Engineering, Tallinn University of Technology, Ehitajate Tee 5, 19086 Tallinn, Estonia; kgprashanth@gmail.com; Tel.: +372-5452-5540; 2CBCMT—School of Mechanical Engineering, Vellore Institute of Technology, Vellore 630014, India

**Keywords:** composites, interpenetrating phase composites, casting, powder metallurgy, metal infiltration

## Abstract

Interpenetrating phase composites are a novel class of heterogeneous structures that have recently gained attention. In these types of composites, one of the phases is topologically continuous and can maintain its structural integrity even if the other phase is removed. These composites are generally fabricated by casting, where the reinforcement penetrates into the precursor matrix as a continuous phase. However, the following dilemma arises: if the same two phases are combined by other powder metallurgical routes (due to differences in the fabrication and interfacial conditions), can they still be called interpenetrating phase composites? The reinforcement is added to the precursor matrix, as in any of the conventional composite processing methods. Most importantly, the reinforcement does not interpenetrate the matrix phase. The present Review discusses the various fabrication routes employed for the fabrication of these interpenetrating phase composites and attempts to identify the correct nomenclature for these composites fabricated via the powder metallurgical approach.

## 1. Introduction

Throughout evolution, humans have dealt with a variety of composite materials (knowingly or unknowingly), like wood (a composite of cellulose and lignin), bone (a composite composed of hard hydroxyapatite blended with soft collagen), and teeth, etc. [1,2,3]. Composites have positioned themselves as versatile materials with applications in every sector that can be named [4,5,6]. Composites are a class of materials combining two or more constituents with different physical/chemical/mechanical/thermal properties [7]. When these individual constituents are combined, they result in materials with unique characteristics that may differ from the properties of the individual components [7]. Traditionally, the main constituents present in composites are the matrix and the reinforcement. The matrix forms the major constituent in the composite (the base material, like the solvent in a solution) and the reinforcement forms the minor constituent (like the solute in a solution) [7]. The reinforcement can take the shape of fibers, laminates, and/or particles (Figure 1). In addition, depending on the type of matrix used, composites may be classified as metal matrix composites (MMC), polymer matrix composites (PMC), or ceramic matrix composites (CMC) [4,7,8]. Similarly, depending on the type of reinforcement added, composites may be called laminar composites (laminates), fiber composites (continuous or discontinuous fibers), or particulate composites (of the same or a different size and distribution) [4,7]. The detailed classification of composites based on the matrix and reinforcement types is shown in Figure 1. The effect of the reinforcement shape, size, and distribution on their macroscopic behavior has been studied extensively for several decades [4,7].

These composites may be combined using several fabrication techniques (casting, powder metallurgy, and/or other forming techniques) [9]. For instance, PMCs can be manufactured using techniques like hand layup, pultrusion resin transfer molding, 3D printing (additive manufacturing), etc. [10,11,12,13]. Similarly, MMCs can be fabricated using stir casting, powder metallurgy (including powder production and a variety of consolidation techniques, like hot pressing, hot rolling, hot extrusion, conventional sintering, spark plasma sintering, etc.), physical vapor deposition, metal injection molding, etc. [14,15]. CMCs are fabricated using powder metallurgical processes (hot consolidation–hot pressing, hot extrusion, sintering, etc.) [16,17,18]. Due to the high melting point of the ceramic matrix, casting-based techniques are not generally employed [19,20,21,22,23]. However, infiltration-based techniques are used to fabricate CMCs, where a fluid with a low melting/boiling point (polymer/metal) infiltrates a porous ceramic precursor/matrix. Such infiltration techniques form the basis of a specific type of composites known as interpenetrating phase composites (IPCs) [24,25].

In MMCs, the addition of ceramic reinforcement can improve their mechanical properties, like their hardness, strength, etc., and their tribological properties if the ceramic reinforcement is added as particles [9,26,27,28,29]. However, the addition of ceramic reinforcements (like SiC, SiN, Al_2_O_3_, etc.) results in the poor bonding of the reinforcement with the matrix at the interface, the agglomeration of the particles, etc. [18,30,31,32]. The addition of these ceramic reinforcements as fibers is superior to the addition of ceramic reinforcements as particles. However, these fibers are unidirectional, leading to anisotropic properties in these composites [1,33,34]. One particular concern with MMCs (especially particle-reinforced MMCs) is their intrinsic toughness. These MMCs (with the addition of reinforcement particles) offer outstanding mechanical properties (including strength, toughness, etc.); hence, a tradeoff between the strength and toughness is required, as in any other metallic material (especially when subjected to strengthening treatments). To overcome such bottlenecks, it is necessary to achieve a strength–ductility tradeoff effect in the design of the composite. With the emergence of advanced digital technologies like artificial intelligence (AI) and machine learning (ML), the design of these MMCs is carried out using digital data science approaches. In this context, both forward prediction with deep learning and inverse exploration approaches employing genetic algorithms have been employed. Such use of novel techniques enhances the application spectrum of these MMCs by improving their properties and the strength–toughness balance. The above suggests that the presence of isolated and discrete reinforcements (Figure 1) may lead to anisotropy in their properties; hence, the continuous dispersion of the reinforcement may be preferred. This has led to the development of the next class of materials in the form of IPCs [35].

## 2. Interpenetrating Phase Composites

Interpenetrating phase composites (IPC) consist of coordinated/unified/integrated phases coexisting to form composites. Scientifically, IPCs are novel class of materials that are heterogeneous in nature, consisting of multiple topologically continuous and 3D interconnected phases that are joined together [6,25,35,36]. Both the matrix and the reinforcement are interpenetrated within each other, as shown in Figure 2 [6]. IPCs are a new class of heterogeneous composites, where novel structural design offers unique characteristics with improved bulk properties [36]. Superior properties (mechanical/tribological/biological, etc.) are observed in these IPCs when compared to the traditional composites with reinforcements in the form of fibers or particles [6,36].

One of the fundamental differences that exists in the fabrication of traditional composites and the novel IPCs is the way in which the matrix and reinforcement are added together. In the fabrication of traditional composites, both these constituents are mixed in the required fractions and are then produced by a variety of conventional casting or powder metallurgical techniques. In IPCs, the matrix and the reinforcement are not mixed during the fabrication process. In this case, firstly, the precursor matrix structure (porous in nature) is fabricated using a variety of techniques, and then the secondary reinforcement phase is added in a continuous fashion [6]. In some cases, the matrix precursor may be also fabricated using a liquid metal dealloying process before the addition of the reinforcement. Since the continuous secondary reinforcement is added at later stages to the matrix, the wettability issue between the matrix and reinforcement is irrelevant during IPC fabrication. The other thermal property, namely the coefficient of thermal expansion of the matrix and reinforcement, plays a key role in IPC fabrication, and in turn, dictates their properties [25]. Since the reinforcement is added at later stages, during the solidification of the reinforcement (in the event that solidification-based techniques are employed), thermal contraction offers a confinement effect at the interface [37]. However, if the reinforcement phase is added externally and the composites are fabricated using the powder metallurgical approach, there will be no confinement effect observed at the interface, unlike fabrication using solidification techniques. In most cases, there will be no interfacial bonding between the matrix and the interface, which also overcomes the formation of brittle intermetallics. Unless an extended time period is offered during which the reinforcement can diffuse with the matrix at the interface (satisfying the conditions of diffusion and solubility), the possibility of an interfacial reaction can be avoided. Since the precursor matrix is fabricated initially, the shape and structure of the matrix can be designed. Similarly, the porosity can be modulated, which in turn determines the structure of the reinforcement or the secondary phase. In addition, the standalone nature of the matrix and the reinforcement offers additional advantages, where they can maintain structural integrity even if either the matrix or reinforcement is removed [36].

Compared to the traditional composites (particle, fiber, or laminar composites), the novel IPCs offer superior out-of-plane strength (both tensile and compressive), ductility, stiffness, and in-plane resilience and toughness [6,36]. Due to the continuous running of the reinforcement, the transfer of stress from the matrix to the reinforcement is facilitated. In addition, the conduction of stress between the matrix and reinforcement is observed to match the performance of natural composites, including biomaterials. IPCs offer a unique toughening strategy where crack propagation can be delayed/arrested. Recently, several reports have been published in the field of IPCs. Most of these IPCs are fabricated with inspiration from nature, where bioinspired 3D-printed Ti-based structures are fabricated and are then infiltrated with biomaterials like Mg-Ti to produce novel next-generation IPCs [24,38,39,40,41,42,43,44]. Even immiscible materials like Cu and W are joined together, where Cu is infiltrated into the W lattice to fabricate W-Cu IPCs for heat-sink-based applications [45]. The applications of IPCs greatly depend on the morphology of the continuous reinforcement, which is based on the fabrication technique employed [36]. These next-generation composites can be used for the following applications: (1) lightweight high-performance applications in aerospace industry, like compressor blades, nose landing gear, ventral fins, rib truss members, axle tubes, drag brace supports, etc.; (2) the automobile industry, like brake pads, drive shafts, piston rods, brake discs, exhaust valves, engine blocks, etc.; (3) thermal management applications, including heat sinks, laser diodes, packaging, transistors, etc.; (4) biomedical applications, including prostheses, implants, etc.; (5) energy applications (capacitors, batteries, etc.); and (6) other miscellaneous applications, including extrusion dies, rollers, erosion-resistant cladding, etc. The different fabrication techniques that can be used to manufacture IPCs are next discussed in detail.

## 3. Fabrication of Interpenetrating Phase Composites

From Figure 2, it is clear that, for the fabrication of IPCs, both a porous precursor matrix and a reinforcement are required. Both the matrix and reinforcement are combined using a variety of fabrication techniques. Firstly, the different methods used to fabricate the matrix (porous precursor) will be discussed, followed by the techniques used to add the secondary phase to the matrix.

### 3.1. Fabrication of Matrix Precursors

Both the matrix and reinforcement form standalone phases in IPCs, so it is crucial to fabricate the matrix phase with the desired open porosity (also continuous in nature) so that the secondary phase can be introduced to form an IPC. Several fabrication techniques may be employed to fabricate the precursor matrix, and these are discussed here in brief.

#### 3.1.1. Freeze Casting

Freeze casting is one of the casting-based manufacturing techniques that is used to fabricate porous precursors. This technique is also called the ice templating process [25]. The freeze casting process involves the preferential nucleation of solids (ice) and their growth, where thermal gradients are defined often in sub-zero temperatures. An aqueous slurry is frozen directionally, and, during this process, crystals grow continuously within the slurry, and the ceramic/metallic particles are rejected at the solid–liquid interface by the progressing solidification front. This process entraps the solid ceramics/metals between the solidified crystals by segregation. After the complete solidification of the slurry, the solidified crystals under sublimation vaporize, leading to the formation of porous ceramic/metallic foams with directionally oriented macropores. These directionally aligned macropores create be a replica of ice. Freeze casting can be used to fabricate a wide variety of open and continuous porous materials, including metals and ceramics. The freeze casting process is eco-friendly since it does not include the burning of material and includes only sintering. However, a disadvantage of this process is the directionality of the pores, leading to direction properties. Ceramic precursors like SiC, TiC, YSZ, Al_2_O_3_, Si_3_N_4_, SiO_2_, etc., can be fabricated using this technique, where water is generally used as the solvent [19,21,22,23,46,47,48,49,50,51,52].

#### 3.1.2. Replica Template

Large, interconnected porous metals/ceramics (with 40–95 vol.% pores) can be fabricated using the replica template method, which is otherwise known as the Schwartzwalder method. Here, the impregnation of the slurry (ceramic/metal) by a porous replica template is employed, where the replica template may be composed of organic sponges. The slurry is maintained between the replica template and the porous structure depending on the interaction between the slurry and the replica template. The slurry layers are subsequently dried at high temperatures, and the organic replica templates are removed carefully. The metallic/ceramic layer may be subsequently sintered to improve its structural integrity. The replica template method is simple, flexible, and versatile. All typical sintering-based defects may be observed in the fabricated porous precursor [53,54,55,56].

#### 3.1.3. Gel Casting Technique

Gel casting is used to fabricate interconnected porous bodies in 3D with the help of slurries. Slurries (composed of ceramics/metals) are fabricated with organic materials (like acrylamides, acrylates, etc.) and cross-linkers. The in-situ polymerization of the organic material leads to the formation of a porous structure. The added organic material may be removed by controlled burning, and sintering may be performed on the remaining porous structure for further consolidation. This technique is very effective for the rapid fabrication of near-net-shaped porous materials, with very high production rates and yields [20,57,58,59,60,61,62,63,64,65,66,67,68,69].

#### 3.1.4. Sacrificial Pore-Forming Agent

Sacrificial pore forming is a widely used methodology that can be applied to fabricate open pores with a high level of porosity. Sacrificial pore-forming agents are used, which are then removed thermally or chemically to form open pores. The pore-forming agents are uniformly distributed like reinforcements in the matrix. They are fabricated like conventional composites, and the added pore-forming agents are then removed carefully by evaporation/sublimation during the curated thermal treatment to form open porous precursors. Several material options can be used as pore-forming agents, like organic waxes, polypropylene, soft metals (like Sn, In, Bi, etc.), and even common salt (sodium carbonate). The nature and type of the pore-forming agent used will determine the pore architecture that will be formed in the precursor [70,71,72,73,74,75,76,77,78].

#### 3.1.5. Additive Manufacturing

Thanks to the advancements of technology, additive manufacturing has been gaining attention for the fabrication of continuous porous structures composed of polymers, metals, and ceramics [79,80,81,82,83,84,85,86,87,88,89]. Since additive manufacturing technologies employ the layer-by-laser manufacturing of materials, the fabrication of structures is made easier using the bottom-up approach. In addition, additive manufacturing technologies, without restriction, can theoretically be used to fabricate any type of structure with any defined shape [90,91,92,93,94,95,96,97,98,99,100,101,102,103]. We will briefly discuss the different additive manufacturing technologies that can be used to fabricate structures.

##### Binder Jetting

Binder jetting is one of the additive manufacturing processes that can be used to fabricate 3D structures with the help of a binding agent. In this process, a thin layer of powder is spread over a substrate plate or on an already deposited layer. A liquid binding agent is added selectively (as per the CAD data), which bonds the powder particles. The layer may or may not be heated to low temperatures selectively (as per the CAD data). The next layer of powder will be deposited, and the iterative process continues until the entire component is built. In this way, a green compact is fabricated, which is subjected to a debinding process to remove/evaporate the binder. The material (after the removal of the binder) is subjected to a sintering process with or without pressure to densify the structure. This process can be used to fabricate metal or ceramic parts without difficulties. This process can also be integrated with the foundry process and will be a good candidate for the fabrication of IPCs [104,105,106,107].

##### Direct Energy Deposition

Direct energy deposition is another additive manufacturing technology that uses a powder or wire as a feedstock material to fabricate 3D structures. The feedstock materials are pushed through a nozzle and are melted by an energy source. The energy source can be a laser, electron beam, plasma, etc. The melt is then directly deposited over a substrate plate or on an already deposited material. The melting and deposition take place continuously, and the nozzle can move in multiple directions at the same time, as directed by the CAD data. The direct energy deposition process is also called the powder-fed fusion process and is similar to traditional welding. However, unlike welding, the direct energy process can be used to fabricate parts with intricate designs. This process is ideal for the fabrication of metallic structures [108,109,110,111].

##### Material Extrusion

Material extrusion is an extrusion-based technology that can fabricate polymer or metallic structures. It is also called the fused deposition modeling (FDM) process. The feedstock material, in the form of a wire, is extruded continuously through a heated nozzle (under constant pressure). The extruded material is then deposited over the substrate as dictated by the CAD data. The processing conditions, like the nozzle temperature, deposition time, etc., are carefully curated to fabricate defect-free parts. Material extrusion is one of the least expensive additive manufacturing technologies available on the market [112,113,114,115].

##### Material Jetting

Material jetting is another additive manufacturing technique, similar to the binder jetting process. However, during material jetting, a photoreactive resin is melted into a liquid photopolymer and is sprayed onto the substrate using a print head. The deposition of the liquid photopolymer can be continuous or on demand. Some of the material jetting processes are the polyjet, nanoparticle jetting, and drop-on-demand methods. The material jetting process offers a high resolution with smooth surfaces [116,117,118,119].

##### Powder-Bed Fusion

Powder-bed fusion is the most widely used additive manufacturing technique, where a powder is used as a feedstock material [120,121,122,123,124,125,126,127]. The powder is deposited over a substrate or an already spread powder bed and is selectively melted by an energy source (laser or electron beam). The substrate is then lowered depending on the layer thickness employed. The next layer of powder is then deposited, and the whole iterative process continues until the entire structure is built. Generally, the process takes place inside a closed chamber, which is either held in a vacuum or flushed with inert gas. This powder-bed fusion process can generally be used to fabricate metallic structures or cermets [128,129,130,131,132,133,134,135,136,137,138,139,140,141]. The different types of powder-bed fusion processes include selective laser sintering, the laser powder-bed fusion process, the electron-beam powder-bed fusion process, direct metal laser sintering, etc. [142,143,144,145,146,147,148,149,150,151,152,153,154,155,156,157,158,159,160,161,162,163,164].

##### Vat Polymerization

One of the first invented additive manufacturing technologies was vat polymerization. Intricate parts with high resolutions (nanoscale) can be fabricated using this technology. Photopolymerization is a chemical process where liquid photopolymer resins are selectively exposed to light (selective wavelength/frequency, e.g., ultraviolet light). During this exposure, a chemical reaction takes place with the resin, and this process takes place layer-by-layer for the building of complex structures. The different types of vat polymerization include stereolithography, digital light processing, and continuous digital light processing [165,166,167].

#### 3.1.6. Liquid Metal Dealloying

Liquid metal dealloying is a wet electrochemical-based approach in which the selective leaching of one of the phases is carried out to create porous scaffolds. The immiscibility gap between the materials added dictates the liquid metal dealloying process. The dealloying process leads to the formation of continuous ligaments. The morphology of the ligament will influence the properties of these porous materials. The pores formed are in the submicron or nano range, and these cannot be fabricated with any of the above-discussed methods. Due to the electrochemical nature of this method, it can lead to the formation of several defects. Lamellar eutectic alloys are one of the best candidates for the dealloying process, since the constituent in the eutectic mixture has no or little solubility at room temperature. In addition, the lamellar structure helps in the easy removal of the secondary phase upon electrochemical attack [35,45,168,169,170,171,172,173,174].

### 3.2. Addition of Secondary Phase to Matrix Precursor

To fabricate IPCs, the secondary phase needs to be added to the precursor matrix with a continuous pore structure. The addition of the secondary phase can be achieved by traditional manufacturing routes, like casting or powder metallurgy (see Figure 3) [36]. The different processing routes will be discussed here briefly.

#### 3.2.1. Casting-Based Approach

Several casting-based methods may be employed to infiltrate the secondary phase into the precursor matrix. This infiltration may take place with the effect of capillary and gravity-driven action (pressureless) or with the application of pressure (forced infiltration). In the case of pressureless infiltration techniques, the capillary action draws the liquid into the preform (the driving force is the capillary action) along with gravity. When pressure is applied during the infiltration process, the external pressure forces the liquid to enter the precursor quickly and efficiently. Some of the most used methods will be discussed briefly below [175,176,177,178,179,180,181].

##### Pressureless Infiltration

Pressureless infiltration (also called gravity infiltration) allows the spontaneous impregnation of the melt into the precursor with the application of any external pressure/force. This process is relatively slower than the other pressure-based infiltration techniques; however, it is relatively more cost-effective. The capillary action plays a significant role, along with the gravitational effects, in allowing the metal to enter the pores of the precursor. The process parameters also play a key role, along with the wettability effects, in this process. If the melt has good wettability with the precursor matrix, this method can be beneficial. Most IPCs are fabricated using the pressureless infiltration technique [182,183,184,185,186,187,188,189,190,191,192].

##### Gas–Pressure Infiltration

The gas–pressure infiltration technique uses a pressurized gas (usually inert in nature) for the effective infiltration of the melt into the porous precursor. The resistance during the infiltration process may be overcome by the pressure exerted by the inert gas. During this process, the melt is allowed to completely cover the porous precursor; subsequently, pressure is applied using a gas flow to infiltrate the melt into the pores. Moreover, in this process, the process parameters, like the temperature of the liquid melt, the applied pressure, the type of gas used, etc., should be carefully optimized. Since pressure is applied, this technique allows the freedom to use a lower melt temperature than in the pressureless infiltration technique. The wettability between the melt and the precursor is no longer an issue, since pressure is applied during the infiltration process, and this process is relatively quicker than the pressureless approach [193,194,195,196,197,198,199,200,201].

##### Squeeze Infiltration

The process of squeeze infiltration (also known as pressure die infiltration) involves the application of very high pressure (50–150 MPa), allowing the rapid infiltration of the melt into the precursor’s pores. The precursor (normally heated) is placed inside a die cavity and the melt is poured over the precursor under a set pressure. The pressure is maintained throughout the process (even during the solidification of the melt inside the precursor). After solidification, the pressure is released, and the IPC is ejected from the die. This process allows the deep penetration of the melt, filling even the micropores present in the precursor. In addition, this method allows the fabrication of IPCs with an intricate design in the second phase. The applied pressure should be carefully modulated for the successful fabrication of IPCs using the squeeze casting technique [193,202,203,204,205].

##### Vacuum Infiltration

The vacuum infiltration technique is a unique fabrication technology where negative suction pressure is used to infiltrate the melt into the precursor matrix. The melt is placed inside a furnace, along with the porous precursor, and then the entire furnace is vacuumed. At this juncture, the melt is in contact with the porous precursor, and, with the application of the negative pressure, the melt infiltrates the precursor. The applied vacuum pressure, the infiltration time, and the temperature of the melt play a critical role during the fabrication of IPCs. This process has low solidification rates, which may lead to an interfacial reaction between the secondary infiltrated phase and the precursor matrix. Additional costs will be incurred since the entire process takes place under a vacuum [206,207,208].

##### Centrifugal Infiltration

The application of centrifugal force is employed for the infiltration of the melt into the porous precursor during the centrifugal infiltration technique. In this method, the porous precursor is placed inside the end of the mold, and the entire setup is rotated, generating sufficient centrifugal force that infiltrates the melt into the precursor. In such processes as centrifugal infiltration, the fluidity of the melt is very important, and it is expected for the metal to have good fluidity. Generally, all cast alloys can be infiltrated using such techniques to fabricate IPCs [176,209,210].

##### Ultrasonic Infiltration

One of the most recent methods introduced for the fabrication of IPCs employs ultrasonic waves (acoustic cavitation bubbles) that stimulate the melt for effective infiltration into the porous precursor. Since bubbles are involved during this process, the burst of such bubbles inside the melt creates enormous pressure, which assists in the effective infiltration of the melt into the precursor’s pores. It is also important to note that ultrasonic waves can help in reducing the contact angle, thereby increasing the wettability between the melt and the precursor. This method can be employed to infiltrate a non-wetting melt into a precursor [211,212,213].

#### 3.2.2. Powder Metallurgy-Based Approach

Powder metallurgy is one of the traditional processes that may be used to combine the porous precursor/matrix with the secondary phase/reinforcement [214,215,216]. When employing powder metallurgical techniques, the secondary phase is added to the precursor matrix (like conventional composite processing) and is then consolidated using appropriate consolidation techniques [216,217,218,219,220,221,222]. The wettability between the secondary phase and the precursor matrix is not a concern during the fabrication of IPCs when using the powder metallurgical approach, since neither the matrix nor the reinforcement will be melted during this process. Several combinations of IPCs are fabricated using the powder metallurgical approach, and the different consolidation techniques that are used to fabricate these IPCs are discussed here briefly.

##### Spark Plasma Sintering

Spark plasma sintering is a pressure-assisted sintering technique, where pressure is applied and, at the same time, the powder is heated with the help of a low-voltage pulsed current. Initially, the powder particles are filled inside the porous precursor and the combination is placed inside the spark plasma sintering die. The application of both thermal energy and pressure allows the consolidation of the powder particles that are introduced inside the porous precursor. The spark plasma sintering process can be used to fabricate a wide variety of IPCs, which may include metals, ceramics, and polymers [217,223,224,225,226,227,228,229].

##### Pressureless Sintering

This technique is the conventional sintering process, where the material is consolidated inside a furnace (with a controlled environment) with the help of thermal energy and without the application of pressure [226,229,230,231,232,233,234]. The secondary phase particles (reinforcement) are filled inside the porous precursor matrix and are placed inside the furnace in an inert atmosphere. The furnace is heated to a high temperature (less than the melting points of both the reinforcement and the precursor) and is held at such a high temperature for a prolonged time, allowing the powder particles to consolidate. The sintering temperature and time play a major role during the sintering process and influence the quality of the product. Since no pressure is involved during the sintering process, the product may have a high degree of porosity when compared to sintering-based processes that involve the application of pressure [235,236,237].

##### Hot Pressing/Hot Extrusion

Another powder metallurgical-based technology used to fabricate IPCs relies on hot consolidation processes like hot pressing or hot extrusion. In this approach, the secondary phase (reinforcement particles) is filled inside the porous precursor matrix and is then hot-pressed or hot-extruded. The hot extrusion process undergoes extensive plastic deformation as compared to hot pressing. The plastic deformation experienced during the fabrication process assists in ensuring good interfacial bonding between the matrix and the continuous secondary phase. It also effectively helps in pore elimination during the consolidation process. The hot pressing/extrusion processes combine two different processes: powder compaction and thermal processing. Hence, employing such hot pressing/extrusion shortens the process flow and hence can reduce the cost of production. Generally, metallic-based IPCs are fabricated using this approach compared to ceramic-based IPCs, since this approach involves considerable plastic deformation and ceramics cannot deform like metals [217,218,219,238,239,240].

## 4. Nomenclature Dilemma

When IPCs are fabricated using the solidification-based casting approach, the secondary phase is infiltrated into the porous precursor with or without the application of pressure. The infiltrated melt interpenetrates the porous precursor to form IPCs (Figure 4). Hence, the term ‘IPC’ is well suited for composites fabricated using infiltration-based casting routes.

On the other hand, if the composites are fabricated using the powder metallurgical approach, neither the infiltration nor the interpenetration of the secondary phase takes place. Instead, the secondary phase is added manually to the precursor matrix, as in the processing of conventional composites. In addition, at present, the discussion is based on the presence of two phases (one being the matrix phase and the other being the reinforcement phase), but the reinforcement is topologically continuous and 3D-interconnected. Both the matrix and the reinforcement are interpenetrated within each other (Figure 2). The situation becomes even more complicated if more than one reinforcement is added to the precursor matrix. For instance, see Figure 5, which illustrates cases where composites with more than two reinforcements are added to the precursor matrix and are consolidated. The precursor matrix shown in Figure 5 is composed of a Fe-rich structure (a honeycomb structure), where three different reinforcements are added, namely Mg, Al, and Ti. In such cases, we cannot call these composites IPCs. Hence, a new scientific term should be introduced to classify or identify these types of composites.

Hence, the present author adopts the following perspective regarding the nomenclature used to refer to these composites.

(a)If the secondary phase is infiltrated into the precursor matrix (generally, via the casting-based approach) so that the secondary phase interpenetrates the matrix, the resultant composite may be called an IPC.(b)On the other hand, if the secondary phase is added to the precursor matrix (the powder metallurgy-based approach), like a conventional composite, the secondary phases are supposed to surround the precursor matrix and do not interpenetrate during the consolidation process; hence, these types of composites may be termed metallic multimaterials (in general) instead of IPCs. These composites may be specifically addressed as metallic bimaterials, since two phases are involved. If both the matrix and reinforcement are metals, the resultant composite may be called a metallic bimetal.(c)If multiple secondary phases are added to the precursor matrix (as in Figure 5), these consolidated samples may be still referred to as metallic multimaterials, since more than two phases are involved in the formation of these composites.

## 5. Conclusions

Recently, interpenetrating phase composites have gained attention due to their flexibility in combining multiple materials (combination of metals, ceramics, and polymers). However, they can be fabricated by both casting and powder metallurgical approaches. It may not be correct to refer to these composites fabricated by the powder metallurgical approach as interpenetrating phase composites. Hence, the present work may be summarized in the following manner.

(1)If the secondary phase is infiltrated into the precursor matrix using a casting-based approach, the resultant composites may be called interpenetrating phase composites, since the reinforcement interpenetrates the precursor matrix during fabrication.(2)On the other hand, if such composites are fabricated using a powder metallurgical approach (hot consolidation, including hot pressing, hot extrusion, etc., or sintering), the composites may be called metallic multimaterials, since the secondary phase is externally added (as in conventional composites manufactured by the powder metallurgical route) and it does not interpenetrate the precursor matrix.(3)If only one reinforcement is added to the precursor matrix, these metallic multimaterials may be termed metallic bimaterials.(4)If both the reinforcement and the matrix are metals, they may be termed metallic bimetals.

## Figures and Tables

**Figure 1 materials-18-00273-f001:**
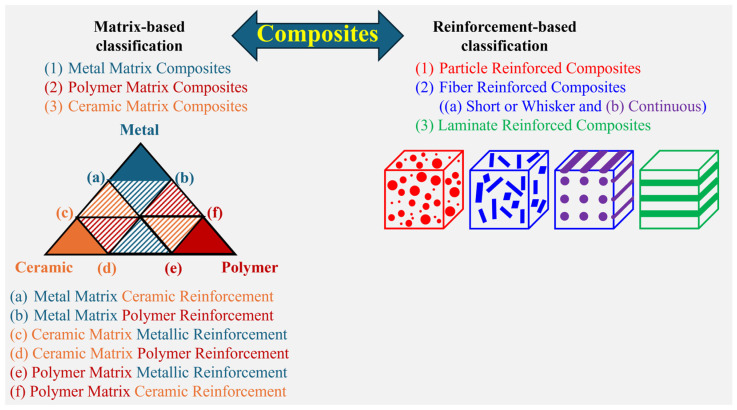
Schematic illustration showing the classification of composites based on both the matrix and reinforcement.

**Figure 2 materials-18-00273-f002:**
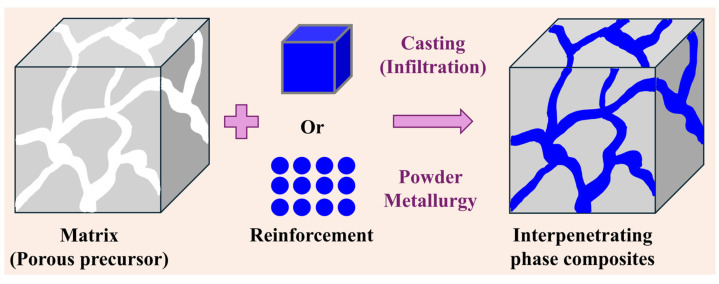
Schematic illustrating the fabrication of interpenetrating phase composites by either casting (infiltration) or powder metallurgical techniques.

**Figure 3 materials-18-00273-f003:**
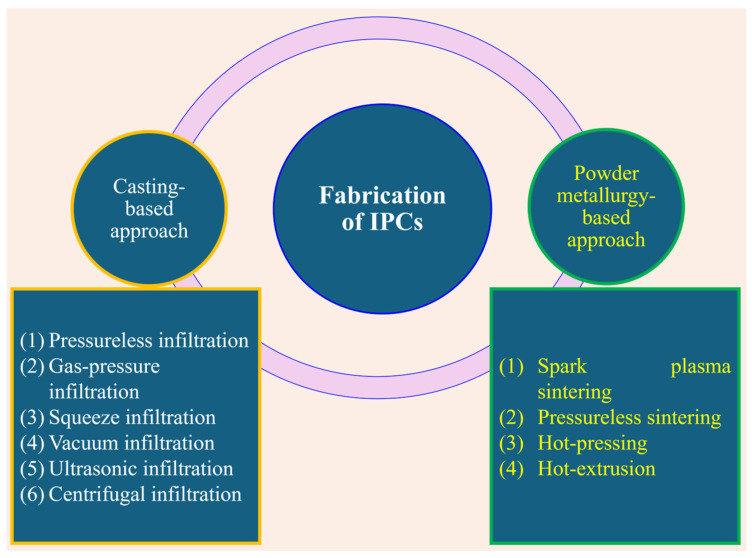
Schematic illustrating different approaches that can be employed to fabricate interpenetrating phase composites.

**Figure 4 materials-18-00273-f004:**
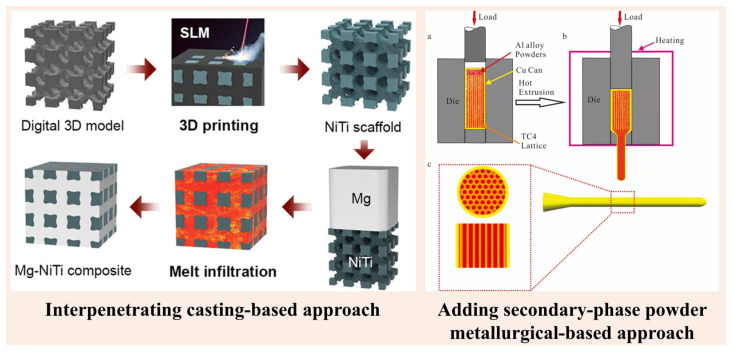
Schematics illustrating the two different approaches that may be employed to fabricate interpenetrating phase composites, namely interpenetration through casting and the addition of a secondary phase through powder metallurgical techniques (figures adapted from [38]—open access article and [241]—master’s thesis). (**a**–**c**) shows the different steps involved during the extrusion process where (**a**) addition of powder to the precursor, (**b**) actual extraction of the composites, and (**c**) extruded MMM composite.

**Figure 5 materials-18-00273-f005:**
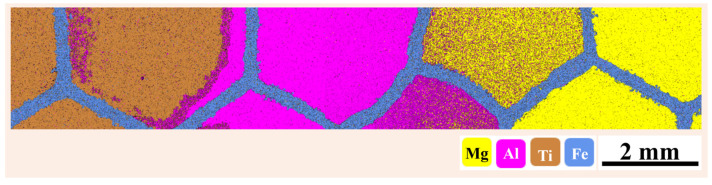
Scanning electron microscopy–energy-dispersive spectroscopy mapping of the elements Mg, Al, Ti, and Fe present in the composite.

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
