# Peer review of "Interpenetrating Composites: A Nomenclature Dilemma"

_materials, 2025, doi:10.3390/ma18020273_

Round 1

Reviewer 1 Report

Comments and Suggestions for Authors

The article "Interpenetrating composites: A nomenclature dilemma"  I believe that can be important to introduce the type of composites that you are writing in the article, are metallic composites?, Please change the title if is suitable

I believe that your article seems a compilation of some techniques of fabrication

Introduce the next points in your article:

Which is the nomenclatures used for composites that you mention in the article, and which is the problem withe nomenclarture in this side.

Which are the main applications or uses of composites cited in your article?

Please rewrite the abstract and conclusions, and indicate the goals of the article, clarificating the goals of the research in introduction

Author Response

Response to Reviewers and Editors comments:

Dear Editor and Reviewers,

We sincerely thank you and the reviewers for conducting the review of the present article and offering us constructive criticism and comments to improve the quality of the present manuscript. Accordingly, we have revised the manuscript, and all the changes are highlighted. Please find the point-by-point response to the comments made.

Reviewer #1:

Q: The article "Interpenetrating composites: A nomenclature dilemma" I believe that can be important to introduce the type of composites that you are writing in the article, are metallic composites?, Please change the title if is suitable.

R: Thanks for your views. I personally feel that the title is apt for this opinion article.

Q: I believe that your article seems a compilation of some techniques of fabrication

Introduce the next points in your article:

Which is the nomenclatures used for composites that you mention in the article, and which is the problem withe nomenclarture in this side.

R: The nomenclature that is presently used is ‘Interpenetration phase composites’ which has been already mentioned. The problem of using the term Interpenetration phase composites has also been mentioned, especially, when these composites are fabricated using powder metallurgical routes. Please refer to section 4: Nomenclature Dilemma.

Q: Which are the main applications or uses of composites cited in your article?

R: The applications of such composites have been introduced in the revised version of the manuscript. The revisions are highlighted suitably in the revised version.

Q: Please rewrite the abstract and conclusions, and indicate the goals of the article, clarificating the goals of the research in introduction

R: As suggested, the abstract and conclusions are modified. This is not a research article and hence there is no goal involved w.r.t. research. On the other hand, this is an opinion article that deals with the present nomenclature and the dilemma involved with the nomenclature, gives clarity, and names an appropriate term for calling this class of composite materials.

Reviewer 2 Report

Comments and Suggestions for Authors

1. In Fig.1, recently, metal particle/fiber reinforced metal matrix composites and ceramic reinforced ceramic matrix composites are emerging, please give some introduction.

2. It is better to give some schematics for every fabrication of matrix precursors method.

3. Please give a comparison and conclusion for each fabrication method.

4. Composite interfaces are very important for interpenetrating composites, please give some details to illustrate them.

5. The conclusion section should be divided several parts, e.g. (1)XXX; (2)XXX; (3)XXX.

Comments on the Quality of English Language

Fine

Author Response

Response to Reviewers and Editors comments:

Dear Editor and Reviewers,

We sincerely thank you and the reviewers for conducting the review of the present article and offering us constructive criticism and comments to improve the quality of the present manuscript. Accordingly, we have revised the manuscript, and all the changes are highlighted. Please find the point-by-point response to the comments made.

Reviewer #2:

Q: In Fig.1, recently, metal particle/fiber reinforced metal matrix composites and ceramic reinforced ceramic matrix composites are emerging, please give some introduction.

R: As suggested, some information about these composites is included.

Q: It is better to give some schematics for every fabrication of matrix precursors method.

R: Thanks for the suggestion. However, the focus of this opinion article is not about the fabrication of matrix precursors to that extent, but also not on the fabrication of interpenetrating phase composites. Hence, I feel it may not be appropriate to include the suggested schematics of all the possible fabrication methods for matrix precursors. I also feel that introducing schematics for every matrix precursor fabricating method will increase the length of the article that may lose its focus. Hence, I have refrained from implementing the suggestion. Sorry.

Q: Please give a comparison and conclusion for each fabrication method.

R: Thanks for the suggestion. However, the focus of this opinion article is not about comparing the different fabrication technologies to produce interpenetrating phase composites. I also feel that the suggested comparison will increase the length of the article that may lose its focus. Hence, I have refrained from implementing the suggestion. Sorry.

Q: Composite interfaces are very important for interpenetrating composites, please give some details to illustrate them.

R: As suggested, some information about the interface is included.

Q: The conclusion section should be divided several parts, e.g. (1)XXX; (2)XXX; (3)XXX.

R: As suggested, the conclusions are numbered (divided into parts).

Reviewer 3 Report

Comments and Suggestions for Authors

Please find my comments in the attached pdf.

Author Response

Pleas find the word file attached
